# Microtubules and Gαo-signaling modulate the preferential secretion of young insulin secretory granules in islet β cells via independent pathways

**Ruiying Hu, Xiaodong Zhu¤, Mingyang Yuan, Kung-Hsien Ho, Irina Kaverina\*, Guoqiang Gu ®\***

Department of Cell and Developmental Biology, The Program of Developmental Biology and the Center for Stem Cell Biology, Vanderbilt University, Nashville, TN, United States of America

¤ Current address: Department of Medicine and Molecular Physiology and Biophysics, Vanderbilt University, Nashville, TN, United States of America

* Guoqiang.Gu@vanderbilt.edu (GG); Irina.Kaverina@vanderbilt.edu (IK)

## Abstract

For sustainable function, each pancreatic islet β cell maintains thousands of insulin secretory granules (SGs) at all times. Glucose stimulation induces the secretion of a small portion of these SGs and simultaneously boosts SG biosynthesis to sustain this stock. The failure of these processes, often induced by sustained high-insulin output, results in type 2 diabetes. Intriguingly, young insulin SGs are more likely secreted during glucose-stimulated insulin secretion (GSIS) for unknown reasons, while older SGs tend to lose releasability and be degraded. Here, we examine the roles of microtubule (MT) and Gαo-signaling in regulating the preferential secretion of young versus old SGs. We show that both MT-destabilization and Gαo inactivation results in more SGs localization near plasma membrane (PM) despite higher levels of GSIS and reduced SG biosynthesis. Intriguingly, MT-destabilization or Gαo-inactivation results in higher secretion probabilities of older SGs, while combining both having additive effects on boosting GSIS. Lastly, Gαo inactivation does not detectably destabilize the β-cell MT network. These findings suggest that Gαo and MT can modulate the preferential release of younger insulin SGs via largely parallel pathways.

## Introduction

In response to postprandial blood-glucose increase, pancreatic islet β cells secrete insulin to promote glucose usage and storage in the liver, fat, and skeletal muscle ensuring blood-glucose homeostasis. The collective β-cell dysfunction, loss-of identity, or death (a.k.a. β-cell failure) results in inadequate insulin secretion [1–3]. This leads to type 2 diabetes (T2D), featured by sustained high blood-glucose levels and deregulated lipid metabolism that damage multiple tissues [4]. In contrast, excessive insulin secretion, caused by cancerous β-cell proliferation [5] or deregulated secretion [6, 7], results in hyperinsulinemic hypoglycemia that leads to comatose or even death.

**Data Availability Statement:** All relevant data are within the manuscript and its Supporting Information files.

**Funding:** This work was supported by grants from National Institutes of Diabetes and Digestive and Kidney Diseases (https://www.niddk.nih.gov/) DK106228 (to I.K. and G.G.), DK125696 and DK065949 (to G.G.), grants from National Institute of General Medical Sciences (https://www.nigms. nih.gov/) R35-GM127098 and R01-GM078373 (to I.K.). K.H. was supported by a postdoctoral fellowship from Eli Lilly and Company (LIFA fellowship 0101420) (https://www.amcp.org/ resource-center/group-resources/residents-fellows/fellowships/Eli-Lilly).

**Competing interests:** One of the co-authors (Dr. Ho) has received a fellowship from Eli Lilly and Company. However, this does not alter our adherence to PLOS ONE policies on sharing data and materials.

Each β cell contains between 5,000 to 13,000 insulin SGs according to ways of morphometric assays [8–12]. According to the physical location of and response to stimulus, these SGs have been traditionally classified into two pools: the readily releasable pool (RRP) and reserve pool (RP) [13]. The former refers to a small group of SGs (<5%) that localize very close to (<20 nanometer) the plasma membrane (PM) [14]. These SGs were immediately released within the first few minutes upon stimulation (i.e. during the first phase of insulin secretion) [14]. In dysfunctional islets from T2D patients, β cells lack this SG pool and the corresponding first phase GSIS [15]. The RRP is absent in newly differentiated immature β cells as well, likely depleted by high levels of basal secretion [16].

The RP contains the majority of insulin SGs in β cells, which are usually located away from the PM. These vesicles need transport to the PM for priming to be released. To this end, high glucose, besides triggering GSIS and new SG biosynthesis, induces the transport of some SGs from RP to replenish RRP [13, 17–20]. These inter-connected responses allow β cells to maintain sustained or pulsatile GSIS under continuous or pulses of glucose stimulation, a property that is necessary for long-term β-cell function.

Intriguingly, not all insulin SGs in the RP are alike and are able to be mobilized to the RRP. In this regard, the half-life of insulin SGs in β cells was reported to be 3–5 days [21]. Several studies, using pulsed radiolabeling or fluorescent-protein tagging of insulin, have shown that young SGs are more likely released upon stimulation [22–27]. Aged SGs will become non-functional and degrade via proteolysis [21, 28]. This degradation ensures long-term β-cell function by removing the non-responsive SGs. It also presents additional metabolic load due to the futile biosynthesis of these SGs, which contributes to the high β-cell stress and reduced cell proliferation/function [29–32].

A feature that potentially contributes to the preferential release of young insulin SGs is their transportability via the microtubule (MT) network. In an elegant study of temporally-marked SGs, Hoboth and colleagues showed that SGs can display three types of glucose-modulated and MT-dependent mobility: highly dynamic, restricted, and nearly immobile states. High glucose can expedite this motion, with the young SGs being more responsive to glucose, and older ones less sensitive and more likely found in the lysosome [33, 34]. These findings support a model that old SGs tend to lose MT-dependent transportability, preventing their movement to the PM and reducing their chance of release.

MTs are cytoskeletal biopolymers that act as tracks for vesicular transport using the kinesin and dynein motor proteins [35, 36]. In many cell types, MTs originate from the centrosome to form a radial array, with their plus ends oriented toward the cell periphery. Henceforth, kinesin- or dynein-mediated transport mediates the bulk flow of cargo toward the cell periphery or interior, respectively [37]. In contrast, most of the β-cell MTs originate from organizing centers in the Golgi [38, 39], centrioles, or endomembrane [40], which form a non-directional meshwork [38–40]. This network is essential for quick/long-range SG movement but is ill suited for bulk directional transport [38, 39, 41–44]. Disrupting [38, 39, 45] or fragmenting [40] these MTs acutely enhances GSIS while stabilizing the MTs represses secretion. These findings imply that β-cell MTs allow active SG-movement between cell interior and cell periphery; however, MTs also compete with the PM for SG binding to acutely reduce the RRP. Without MTs, the overall long-range SG transport is reduced [39, 43, 44]. Yet the non-MT-dependent SG movement, within a brief period of time, is sufficient for a portion of SGs to move to the PM for regulated release [39]. Thus, regulating the dynamics and density of MTs will likely influence the releasability of young versus old SGs, because the new SGs will lose their advantage of being moved to cell periphery when MT-aided transport is reduced.

In addition to transport, SG association with the PM is another limiting step for insulin secretion [18, 20, 46, 47]. In this case, vesicular and PM proteins form a SNARE complex via

the association between Synaptobrevins, Syntaxins, SNAP23/25, Munc18, Rim, and others [48]. This complex brings the vesicles close to the PM. The presence of $Ca^{2+}$, via a family of $Ca^{2+}$ sensors such as Synaptotagmins (Syts) [16, 49, 50] and/or Doc2B [51–53], further modulate the conformation of the SNARE-complex to enable vesicular/PM fusion [54]. Thus, mutations in several of these SNARE components were found to deregulate SG association with PM and subsequent secretion [20, 47, 55–58].

An intriguing molecule that can potentially regulate both SG localization and MT dynamics is inhibitory G protein Gαo. Gαo signaling toggles between on- and off-state by dissociating/associating with Gβγ dimers in response to the activation of G-protein coupled receptors [59]. Unlike other inhibitory Gα (Gαi1, Gαi2, Gαi3, and Gαz), Gαo activation does not inhibit adenylyl cyclases in several cell types including islet β cells [18, 60], but repressing GSIS by reducing SG location close to the PM [18]. Intriguingly, Gαo at high levels, together with other inhibitory Gα subunits, was shown to promote MT disassembly by Gα-MT association [61, 62]. Here, we explore the hypothesis that Gαo may regulate SG transport through the MT network, which consequently control the probability of SG secretion in young versus old SG pools.

## Results and discussion

### The β-cell MTs are dispensable for overall SG-distribution near β-cell membrane

We have previously shown that the MT network in β cells, although essential for sustained secretory function [45], acutely represses GSIS [39, 45, 63]. Based on Total Internal Reflection Fluorescence Microscopy (TIRFM)-observation of MT-dependent insulin SG movement near cell membrane (<200 nm away from the PM), we postulated that a role of the MT network in β cells is to pull SGs away from the PM besides acting as tracks for their long-distance movement [33, 38, 39]. Here, we corroborated this hypothesis by first examining the overall distribution of insulin SGs in β-cell cytoplasm that is close to or away from the PM with or without MTs.

Isolated islets were stained for insulin immunofluorescence (IF) and analyzed via confocal microscopy after incubation in Nocodazole (NOC) for 12 hours, which effectively destabilized MTs (Fig 1A) while enhancing GSIS compared with DMSO-treated controls (Fig 1B). Note that only a small portion of insulin SGs shows fast long-range MT-dependent movement [33, 38, 39, 41]. We therefore used this 12-hour-accumulation to increase the power of detecting changed insulin SG localization.

Insulin IF levels were examined at different areas of β cells (Fig 1C–1F), subdivided into zones that are one-micrometer within and away from PM (Fig 1E and 1F). At both non-stimulating 2.8 mM glucose (G2.8) and stimulating G20, there were significant reductions in total SG levels when cells were treated with NOC (Fig 1G). More importantly, the portion of insulin signals near the PM is significantly higher in (G20 + NOC)-treated samples than the controls (G20 + DMSO), although this difference is not significant at G5.6 (Fig 1H).

At least two known factors can contribute to the enriched insulin signals near the PM of (G20 +NOC)–treated β cells. First, MT-destabilization will inhibit insulin biosynthesis, contributing to lower levels of insulin in the center of NOC-treated β cells [45]. Second, the absence of MTs will abolish the MT-dependent SG retrograde movement of SGs to lysosomes for degradation. Yet these factors, individually or in combination, cannot account for our overall observation, because we detected higher portions of insulin localization near the PM of NOC-treated cells that continue to secrete higher levels of insulin than controls (Fig 1B), while SG-degradation in general is a slow process [28]. Thus, a parsimonious explanation is that

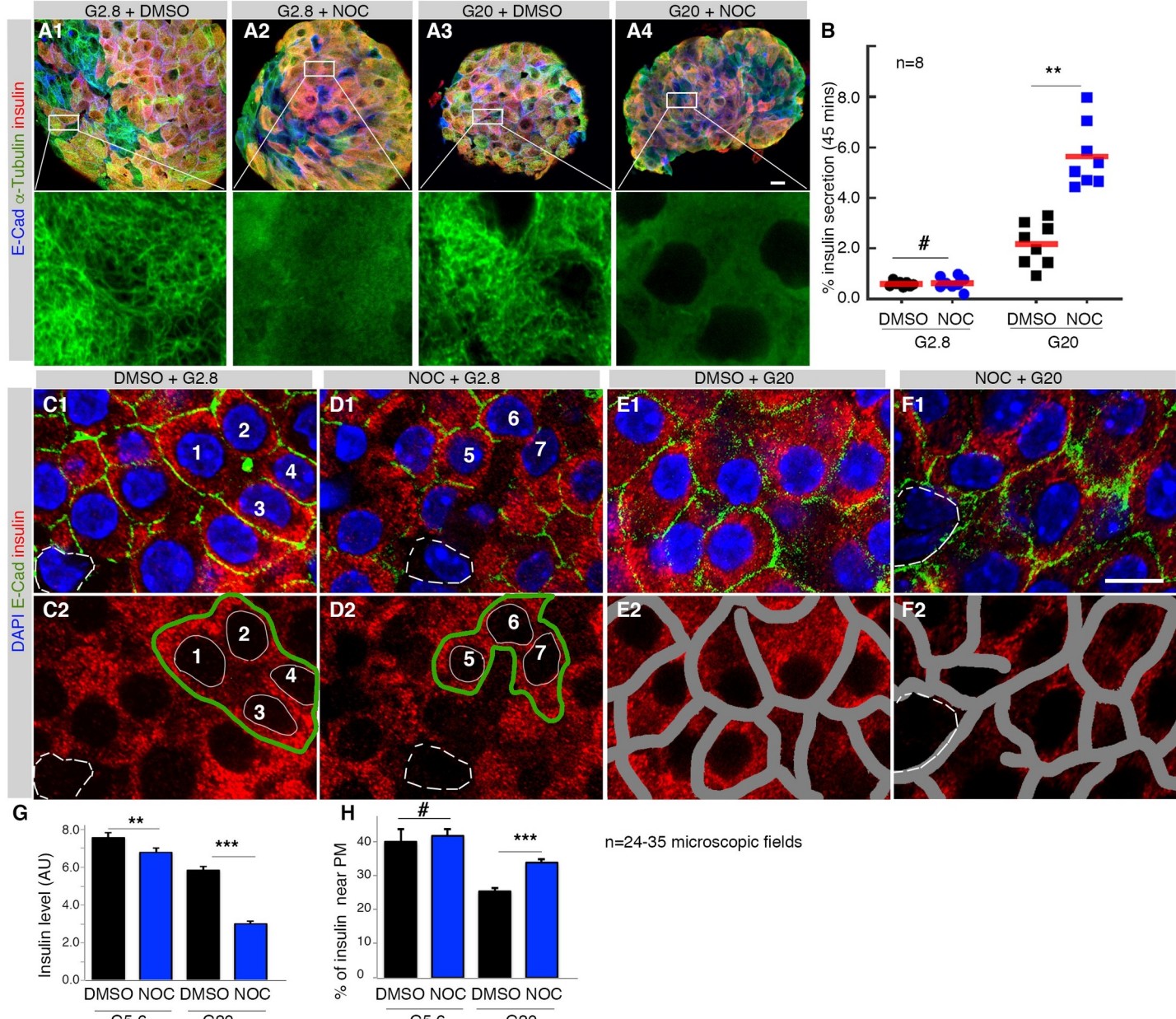

**Fig 1. MTs are dispensable for insulin SGs localization near β-cell periphery.** (A) MT-destabilization by NOC-treatment [A1, 2.8 mM glucose (G2.8) plus DMSO; A2, G2.8 + NOC; A3, (G20 + DMSO); A4, (G20 + NOC)]. Images shown on the top row are maxi-projections of whole mount stained islets, showing co-staining of insulin (red), E-cadherin (E-Cad, blue), and α-tubulin (green). The bottom row includes high-magnification of boxed areas of the top panels, highlighting MT-destruction by NOC. (B) GSIS in islets treated with NOC for 12-hours, scatter plots with mean shown. (C-F) Insulin–subcellular localization (red) in β cells. In (C1-F1), DAPI and Ecadherin signals locate nuclei and cell membrane. A few Ins-negative cells were marked in C, D, and F (white broken circles). (C2, D2) The scheme to assay relative insulin levels in β cells: clusters of β cells showing clear nuclear signals were selected [green circles in C2 (cells 1–4), D2 (cells 5–7)] after excluding the nuclear areas, white circle. (E2, F2) The scheme to examine subcellular insulin localization. B-cell zones along the PM (±1 micrometer, grey bands) were marked and the portion of insulin signals that localize within these zones were determined. (G, H) Relative insulin levels and the portions of insulin that localized near PM, shown as mean + SEM. In (B), (G), (H), #: p>0.12, **: p<0.01, ***: p<0.001, calculated using multi-comparison ANOVA. Please refer to S1 Table for original numbers used in plots.

insulin SGs can move to near the PM in the absence of MTs, where SG–PM associations (e.g., via SNARE complex formation) or other unknown mechanisms can trap these SGs near the PM, which facilitates their stimulated secretion. In order to corroborate this latter possibility,

we utilized transmission electron microscopy (TEM) to directly examine the number of insulin SGs that have very close contact (<10 nanometer away) with β-cell PM.

MT-depolymerization significantly increased the number of SGs that localize within 10 nanometer of β-cell PM at basal glucose (Fig 2A–2C). In contrast, there is a significant decrease in the density of SGs in β cells without MTs (Fig 2D–2F). Similarly, an increased number of SGs close to PM was observed in the absence of MTs under high glucose conditions (Fig 2G–2I), despite significant degranulation with or without MTs under high glucose (Fig 2J. Compare with Fig 2F).

Overall, the above confocal and TEM results are consistent with a model that MTs compete with the PM for insulin SG binding besides their classical roles in SG transport. Specifically, SGs likely associate with MTs via vesicle-bound motor proteins. This allows the MT-dependent SG-transport from the site of biogenesis, the trans-Golgi network that usually localizes in interior of cells, to close-to the PM [38, 41–44, 64, 65]. However, the β-cell MTs have no obvious directionality. Thus, the MTs can also pull SGs away from the PM to prevent their association to PM. When MTs are destabilized or fragmented near the PM, e.g., in the presence of high glucose or NOC, the SGs can lose MT contact and be available for PM-association/secretion. Consequently, SGs with preferential binding with MTs, especially those young ones [33], are more likely transported to the cell periphery for release [63]. The older vesicles, less likely transported due to their attenuated MT-association, will eventually be degraded. Note that all SGs that localized close to PM are not secreted during stimulation. Thus, it is also possible that a portion of these non-secretable SGs depends on MTs to return to β-cell interior for degradation. Our current studies cannot address this possibility.

## Disrupting the MTs allows increased secretion of older insulin SGs

The above model is consistent with the findings that when the MT network is disrupted, the fast SG movement will be abolished [33, 39, 41]. A further prediction is that vesicle movement [slowed but still detectable in the absence of MTs [39]] via free-diffusion or actin-assisted transport is probably sufficient for a portion of SGs to move to close to the PM for secretion [62, 66]. In this setting, older SGs can be mobilized to allow enhanced secretion without additional SG biosynthesis. In addition, the young SGs with superior MT-dependent transportability will lose the advantage of being moved to underneath the PM [33]. In other words, the absence of MTs could increase the secretion probability of old SGs, which can account for the enhanced GSIS in NOC-treated β cells without relying on increased new SG biosynthesis.

GSIS was carried out in the absence of MT and acute protein biosynthesis (Fig 3A–3E). Islets were incubated in the presence of 10 μM cycloheximide (CHX) and 10 mM glucose (G10) for three-hours, a condition that effectively inhibit protein biosynthesis in islets [67]. Treated islets were then incubated with 10 μg/ml NOC and G10 for one additional hour to disrupt islet-cell MTs (Fig 3A–3D), followed by protein synthesis assessment (Fig 3E) and GSIS assays (Fig 3F). Because newly synthesized insulin become secretable within two-hours [68], we expect this treatment to reduce newly produced secretable SGs. The three-hour incubation was chosen because it allows sufficient number of new SG-biosynthesis in control β cells for comparison. The effects of reduced production of other secretory proteins on GSIS were controlled by including both DMSO- and NOC-treated islets [67]. The lack of protein biosynthesis (Fig 3E) results in a strong trend of reduced GSIS in the presence of MT (i.e., without NOC-treatment, p = 0.066, Fig 3F columns 2 and 6). Yet this lack of protein synthesis did not eliminate the MT-destabilization-enhanced GSIS (Fig 3F, note the significant differences between columns 2 and 4 and between columns 6 and 8). These results are consistent with a conclusion that MT-destabilization does not depend on newly synthesized SGs for enhanced secretion.

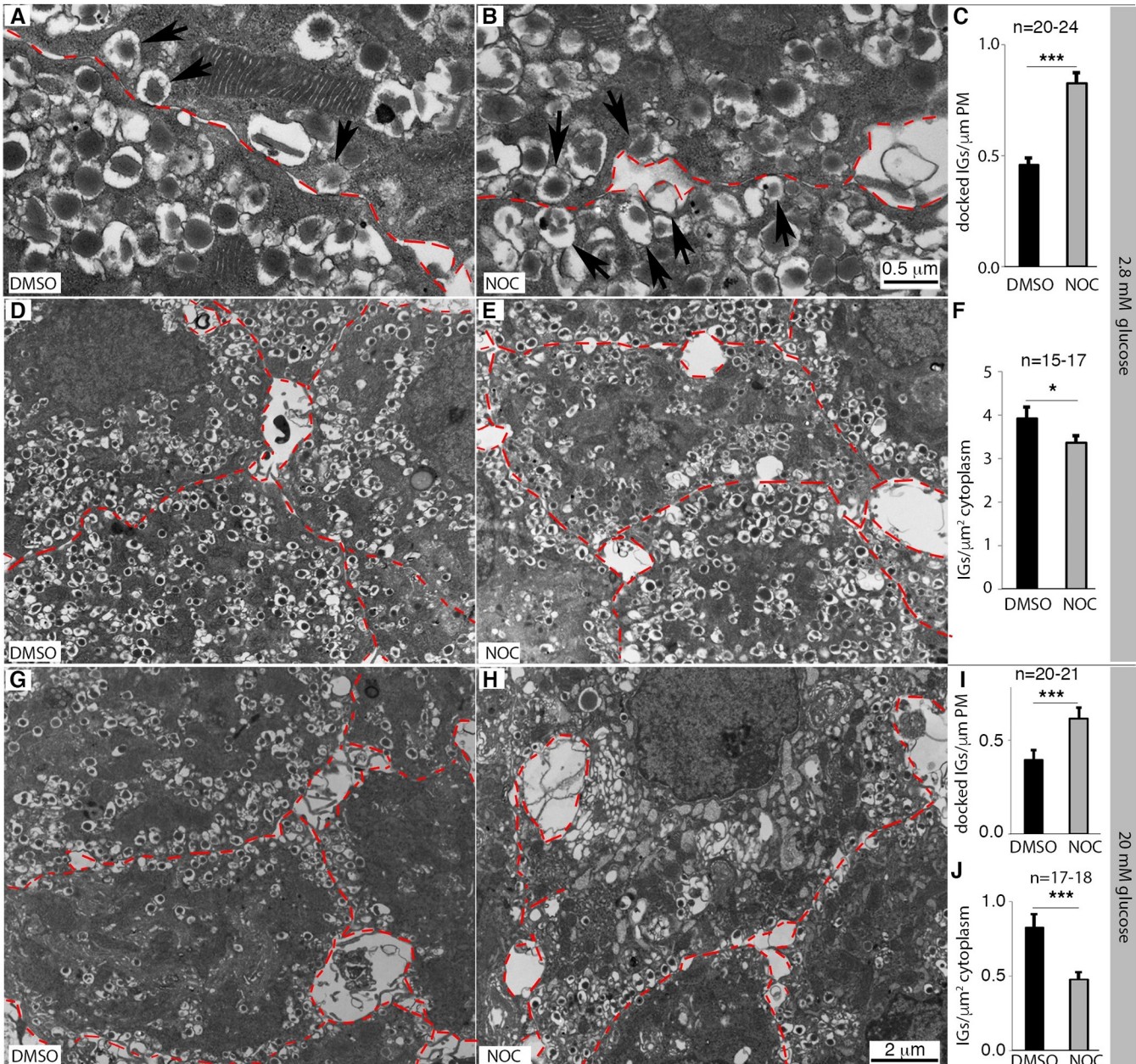

**Fig 2. TEM assay of insulin SG in β cells in the absence of MTs.** Islets were isolated from wild-type (WT) adult mice and were treated with DMSO or 10 μg/ml NOC for ~12 hour at G2.8 or G20 mM. TEM was used to examine the locations and density of SGs in two batches of islets prepared on different days. (A-C) TEM images and quantification of SGs that localize near PM from DMSO- and NOC-treated islets in the presence of 2.8 mM glucose. (D-F). Images and quantification of SG density in microscopic fields used in (A-C). (G-J) Images and SG quantification as in (A-F), except 20 mM glucose was used. Scales in (D, E, G, H) are the same, labeled in panel (H). In (C), (F), (I), and (J), mean + SEM were presented ($^*$: p<0.05, $^{***}$: p<0.001, from two-tailed type II t-test). In all panels, "n" indicates the number of microscopic fields counted (with 3–4 different β cells included in each field). In all images, red dashed lines mark recognizable β cell membrane. Please refer to S1 Table for original numbers used in plots.

Because each GSIS process only releases a small portion of the SG storage, one argument is that a three-hour CHX-treatment cannot appreciably reduce the amount of secretable insulin. Thus, studies described in Fig 3F are not sensitive enough to test if MT destabilization-enhanced GSIS depends on newly synthesized insulin SGs. To address this issue, we compared the secretion-probability of young and old insulin SGs directly without perturbing protein

biosynthesis. Isolated islets were incubated for four-hours with $^{35}$S-labeled Cys and Met. Note that we have included non-radioactive Cys and Met (at two-third concentration of RPMI 1640 media) in the labeling mix, which allows us to avoid potential issues with Cys/Met depletion.

Labeled islet was used for GSIS assays after two-hours of chase in normal media with/without NOC treatment, with NOC disrupting MTs. MT-disruption islets secrete insulin consists of a smaller proportion of radioactive (i.e. younger) molecules (Fig 3G), while secreting more total insulin (Fig 3H). These data suggest that MT-disruption has allowed more old SGs to be secreted, supporting our model that the dense MT network in β cells traps old SGs and prevents their secretion during GSIS. Without MTs, a bigger portion of old SGs will be made available for GSIS, leading us to explore the mechanisms that regulate both the MT networks and older vesicle secretion in β cells. Note that we detected a higher portion of younger SG release at basal glucose (Fig 3G). The exact reason for this finding is not clear. It is possible that young SGs are heterogeneous and having different sensitivity to glucose/Ca$^{2+}$. A basal condition favors the release of young SGs with higher sensitivity, with the rest stored and aged for stimulated release, a phenomena that occurs during β-cell maturation [16]. Alternatively, the DMSO that is necessary to dissolve NOC may change the PM property, causing non-physiological secretary properties.

## Gαo inactivation preferentially increases the secretion-probability of old insulin SGs

The increased SG localization close the PM in MT-destabilized β cells is similar to what we have observed in the pancreatic specific *Gαo* mutant (*Gαo$^{F/F}$; Pdx1$^{Cre}$*) mouse β cells [18]. Based on the

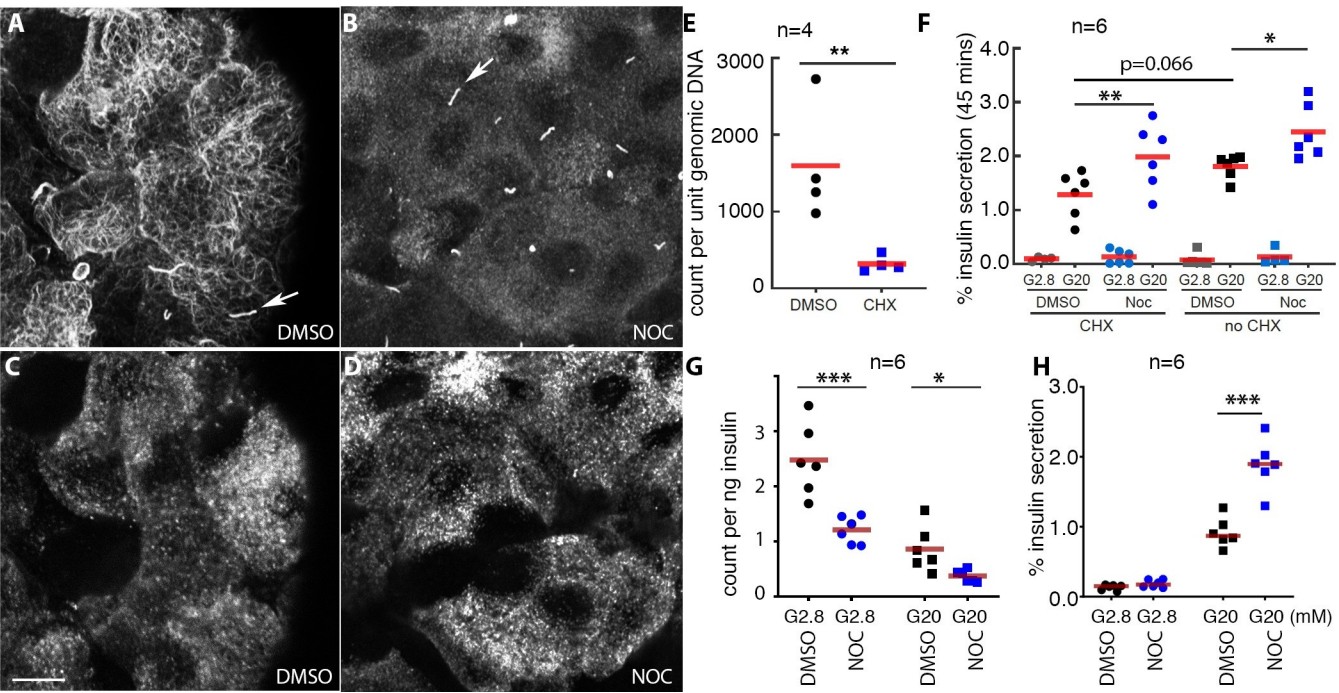

**Fig 3. Newly synthesized SGs are dispensable for MT-destabilization-enhanced GSIS.** (A—D) Glu-tubulin was stained to verify the effective disruption of MTs by NOC (A, B), with β cells identified by insulin staining (C, D, showing the corresponding fields in A, B, respectively). Arrows in (A, B), primary cilia. Scale bar = 10 μm. (E) Total radioactive amino acid (3H-Leu/Ile) incorporation after 3-hours CHX-treatment. The radioactivity was normalized against DNA levels, compared via real-time PCR. Also see S1 Fig for unclipped western images. (F) GSIS from islets treated with combination of 10 μg/ml NOC and 10 μM CHX. Note that the presence of CHX did not eliminate the NOC-potentiated GSIS (compare columns 2 and 4). (G) The radioactive insulin that were secreted (count/ng insulin) in control and NOC-treated islets, following a 4-hour radiolabeling process. (H) % of total insulin secretion in samples used in panel (G). In (E-H) *: p<0.05; **: p<0.01; ***: p<0.001, values from Holm-Sidak's multiple comparisons (E, G, H) or Turkey's multiple comparisons (F). The p-values, not shown between low-glucose samples, are all above 0.3. Please refer to S1 Table for original numbers used in plots.

published findings that trimeric G proteins can regulate MT dynamics [61, 69, 70], we explored the possibility that Gαo regulates SG secretion through MTs. We first tested if Gαo inactivation would impact the preferential secretion of young SGs. $Gαo^{F/F}$; $Pdx1^{Cre}$ β cells, wherein $Gαo$ is efficiently inactivated in β cells (Fig 4A–4E), secrete a larger portion of older vesicles under high-glucose stimulation (Fig 4F). Note that we did not observe higher level of young SG secretion at basal glucose in $Gαo^{F/F}$; $Pdx1^{Cre}$ β cells as in NOC-treated islets. The reason for this is not known but likely related to the properties of SGs and/or lack of DMSO usage (see below).

## Gαo inactivation does not significantly alter MT stability and density

Both Gαo inactivation (in $Gαo^{F/F}$; $Pdx1^{Cre}$ mice) and MT-destabilization can potentiate GSIS [18, 39]. We therefore tested if these two processes depend on mobilizing a same pool of

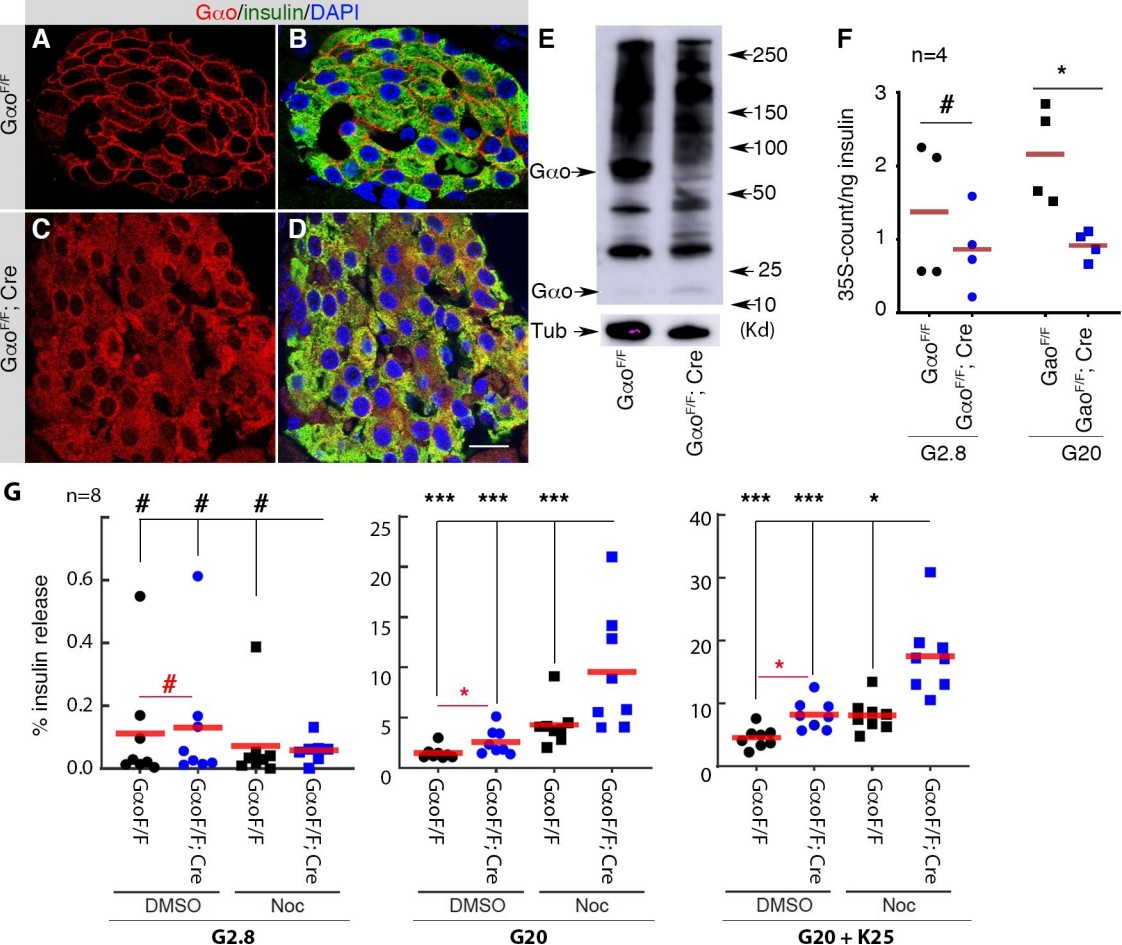

**Fig 4. Gαo and MT regulate SG secretion via parallel pathways.** (A-E) Immunofluorescence and western blot showing Gαo inactivation in $Gαo^{F/F}$; $Pdx1^{Cre}$ islet cells. Note that Cre-mediated $Gαo^F$ deletion will yield a mRNA that translates a short N-terminal Gαo peptide, recognized by the antibody (red) but has no detectable biological effect [18]. Full length funtional Gαo is membrane-bound, while the N-terminal fragment is cytoplasmic, allowing for ready verification of $Pdx1^{Cre}$-mediated $Gαo$ inactivation in insulin + (green) cells. DAPI (blue) stained for nuclei. Scale bar, 20 μm. In (E), note the disapearance of Gαo full length protein and the appearance of a short Gαo fragment in the $Gαo^{F/F}$; $Pdx1^{Cre}$ islet samples. Alpha-tubulin was used as loading controls for the western blot. (F) The levels of $^{35}$S-labeling in secreted insulin from control and *: p<0.05; **: p<0.01; ***: p<0.001)β cells, prelabeled for four hours. (G) Insulin secretion from $Gαo^{F/F}$; $Pdx1^{Cre}$ islets, with or without NOC-treatment, induced by basal G2.8, stimulating G20, and KCl-induced depolarization [(A, B) G20 + K25 (25 mM KCl)]. P values, results from multi-comparison ANOVA or one-way ANOVA (#: p>0.3. *: p<0.05; **: p<0.01; ***: p<0.001). Note the red font "#" or "*" in (G) highlight the differences between control and $Gαo^{F/F}$; $Pdx1^{Cre}$ islets. Please refer to S1 Table for original numbers used in plots.

insulin SGs for enhanced GSIS. If so, we expect that MT disassembly in $G\alpha o^{F/F}$; $Pdx1^{Cre}$ β cells will not further increase GSIS. In contrast, treating $G\alpha o^{F/F}$; $Pdx1^{Cre}$ β cells with NOC induced an additional enhancement in GSIS (Fig 4G). These data are consistent with a model that Gαo inactivation and MT-disassembly facilitate the release a different sub-pools of insulin SGs.

## Gαo activity does not stabilize MTs in β cells

We finally tested if Gαo regulates MTs by comparing the MT density and stability in control and $G\alpha o^{F/F}$; $Pdx1^{Cre}$ β cells. No difference in MT density (stained for tubulin) was observed between control and $G\alpha o^{F/F}$; $Pdx1^{Cre}$ β cells when examined in single cells (Fig 5A and 5C) or whole islets (Fig 5B and 5D), as measured by the average distances between MT filaments (Fig 5E). Similarly, we did not observe significant differences in MT-stability in $G\alpha o^{F/F}$; $Pdx1^{Cre}$ and control β cells when the levels of Glu-tubulin [a well-established marker for MT stability,

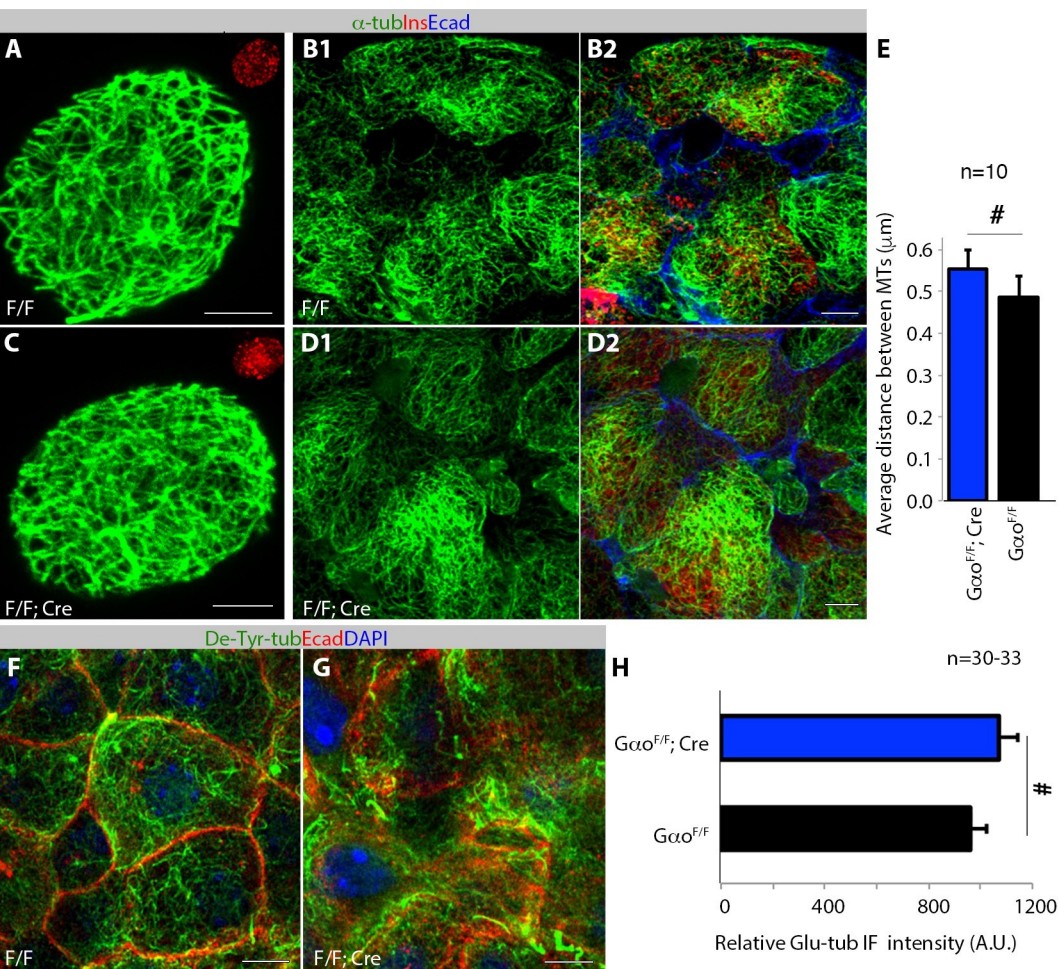

**Fig 5. Inactivating *Gαo* does not alter MT stability and density in β cells.** (A-E) The MT density in control and *Gαo^{F/F}*; *Pdx1^{Cre}* β cells, stained for tubulin (green), insulin (red), and E-cadherin (blue). Single cells attacehd to coverslips (A, C) or intact islets (B, D) were used. Insets in (A), (C), insulin staining to identify β-cells. Note that in (B, D), a single α-tubulin and a composite channel are presetented. The quantification data in E is MT density in single β cells, which measured the average distances between microtubules at any raodom direction. (F-H) The density of Glu-tubulin in control and *Gαo^{F/F}*; *Pdx1^{Cre}* β cells. Glu-tubulin (green), E-Cadherin (red), and DAPI (blue) signals via immunofluorescence are shown. Presented data in (E) and (H) are (mean + SEM). #: p>0.2, results from two-tail type II t-test. Scale bars, 5 μm. Please refer to S1 Table for original numbers used in plots.

resulting from the removal of the C-terminal tyrosine in α tubulin to expose the glutamate residue [63]] were compared (Fig 5F–5H). These findings suggest that inactivating Gαo does not alter MT dynamics in β cells. Note that there are weak trends of increased MT-density (p = 0.31) or MT-stability (p = 0.24), in $Gαo^{F/F}$; $Pdx1^{Cre}$ β-cells compared with controls (Fig 5E and 5H). However, because both Gαo-inactivation and MT-destabilization enhance GSIS, it is unlikley that this weak MT-regulation by Gαo plays key roles in GSIS.

In summary, we revealed two factors that potentially act in parallel to impact the secretion probability of young versus old SGs in islet β cells. One is the non-directional MT meshwork. A model is that the MTs act as holding places for SGs in the RP, whose transition to the RRP was expedited by MT depolymerization or fragmentation and transport by kinesin and dynein motor proteins [41, 43, 44]. This mechanism is advantageous in that it prevents insulin over-secretion. However, when SGs age and lose their association with MTs, they will no longer be useful for function in the presence of a dense mesh of MTs and therefore must be degraded. This poises extra stress for β cells to replace this stock via insulin biosynthesis [71, 72]. Note that although the absence of MTs can improve the usage of the old vesicles, β cells without MTs cannot maintain long-term function, due to the essential roles of MTs in β cells for new insulin biosynthesis [45].

The other factor that reduces the probability of secretion of old insulin SGs is Gαo, whose inactivation enhances GSIS by increasing release of older SGs as with MT-destabilization [18]. Yet, Gαo does not play large roles in regulating MT stability or density in β cells as predicted based on *in vitro* biochemical studies [73, 74]. An interesting future investigation could be testing if Gαo can regulate the affinity/processivity of motor proteins along the MTs. Related with this possibility, Gαo has been reported to localize to vesicle membrane in chromaffin cells [75, 76], which may impact vesicular transport and/or releasability directly. Moreover, Gαo has been reported to impact the β-cell actin network [75, 76]. Given the important roles of F-actin in GSIS regulation [42], another interesting future work is to examine the chemical/functional interaction between Gαo and β-cell actin. It would also be worthy to examine if Gαo regulates the affinity between SG and PM components that form the SNARE complex. This latter possibility is particularly attractive because Gαo has been shown to interact with Syntaxins [77], limiting factors for SG-association with PM in β cells [56].

## Research design and methods

### Ethics statement and mouse usage

The Vanderbilt Institutional Animal Care and Use Committee specifically approved this study (M18000195 for GG/IK). Mice were euthanized by isoflurane inhalation in compliance with AAALAC policies. Wild type CD-1 (ICR) mice were from Charles River (Wilmington, MA). $Gαo^{F/F}$ and $Pdx1^{Cre}$ mice were described in [18].

### Islet isolation and routine GSIS

Islets were isolated from 8–16 week-old mice using collagenase perfusion as in [29]. Briefly, ~2ml of 0.5 mg/mL of collagenase IV (Sigma, St. Louis, MI) dissolved in Hank's Balanced Salt Solution (Corning, Corning, NY) was injected into the pancreas through the main duct. The pancreas was digested at 37˚C for 20 minutes and washed 4 times with [RPMI-1640 media with 5.6 mM glucose (Gibco, Dublin, Ireland) + 10% heat inactivated fetal bovine serum (FBS, Atlanta Biologicals, Flowery Branch, GA)]. Islets were handpicked and let recover at 37˚C in the same media for at least 2 hours before down-stream experimentation.

GSIS follows routine procedures. Briefly, islets were washed twice with basal KRB solution (111 mM NaCl, 4.8 mM KC, 1.2 mM $MgSO_4$, 1.2 mM $KH_2PO_4$, 25 mM $NaHCO_3$, 10 mM

HEPES, 2.8 mM glucose, 2.3 mM $CaCl_2$, and 0.2% BSA). Islets were then incubated in the same solution (37˚C) for one hour, changed to new KRB to start the secretion assay. For insulin secretion induction, glucose (0.5 M) and/or KCl (1M) stock solutions were directly added to the KRB to desired concentrations. The secretion period assayed lasts 45 minutes. After secretion, islets were immediately frozen-thawed twice between -80˚C and room temperature. Acid alcohol extraction (70% alcohol + 0.2% HCl) was then performed at 4˚C overnight to determine the total insulin content. For each GSIS assays, 8–15 islets were used in 1 ml KRB. The insulin levels were then assayed using an ultrasensitive Insulin Elisa kit from Alpco after dilution to within the range of sensitivity.

## Islet pretreatment–MT destabilization, radiolabeling for protein synthesis inhibition, and radiolabeling

For short-term NOC treatment, islets were pre-incubated in KRB (2.8 mM glucose) with 10 µg/ml NOC (with 10 mg/ml stock in DMSO) for one-hour to depolymerize MTs. This level of NOC was included in all solutions afterwards. DMSO (<0.05%) treated islets were used as controls. For overnight NOC-treatment, RPMI1640 media with G2.8 or G20 were used.

To inhibit protein translation, islets were incubated in RPMI-640 with 10 mM glucose and 10 µM CHX for three hours to minimize the reduction of other proteins that are essential for secretion [67]. Secretion assays follow the above procedure.

For protein radiolabeling, isolated islets were incubated for 3-hours in RPMI-1640 [supplemented with 10% FBS, 10 mM glucose, and 1/30 volume of $^3$H-labeled leonine/isoleucine (#NET1166001MC, Perkin Elmer)] cultured at 37˚C with 5% $CO_2$. Islets were then washed three times and chased in the same media without radioactive amino acids for 2 hours. Islets were then lysed using cell lysis buffer [20 mM Tris (pH: 8.0) + 150 mM NaCl + 0.2% SDS + protease inhibitor cocktail (ThermoFisher, A32963)], sonicated, precipitated with Tri-chloric acid to read the total $^3$H-incorporation. The reads were then normalized against the cell number for comparison. For this goal, a portion of the lysate was saved to assay the number of cells via real-time PCR that assays the copy number of *Ngn3* locus (oligos used are 5′-GCGCAAG AAGGCCAATGA-3′ and 5′-CAGCGCCGAGTTGAGGTT-3′).

For secretion assays of young SGs, islets were incubated for 4-hours in a mix of RPMI-1640 and Met/Cys-free DMEM media (3:1 ratio) [supplemented with 10% FBS, 10 mM glucose, and 1/30 volume of $^{35}$S-labeled Met/Cys (#NEG772002MC, Perkin Elmer)] cultured at 37˚C with 5% $CO_2$. Islets were then washed three times and chased in cold RPMI-1640 media for 2 hours and proceed to GSIS and radioactivity assays. Briefly, the insulin levels were quantified using an Elisa kit. The radioactivity in secreted insulin was quantified using a scintillation counter (Beckman LS System 6000TA) following immunoprecipitation (IP) using guinea pig anti-insulin (Dako, Santa Clara, CA, #A0564) and Protein-A beads (ThermoFisher). For each IP assays, 2 µl antibodies and 30 µl Protein-A beads were used.

## Transmission electron microscopy (TEM) detection of SGs

Islets were incubated in RPMI-1640 with 10% FBS plus 5.6 or 20 mM glucose with or without 10 µg/ml NOC for 12 hours. Islets were then fixed, sectioned, and imaged following routine TEM protocols as detailed in [16]. To count the number of SGs associating with PM, Image J was used to measure the length of β-cell membrane. The vesicles with near direct contact with the PMs (<10 nm away) were counted. The SG density was counted in a similar fashion, except that β-cell cytoplasmic areas were selected and measured. Double-blind tests were used without identifying the treatment conditions first.

## Immunofluorescence (IF), western blot, and microscopy

For Gαo staining, routine frozen pancreatic sections were used [29]. Briefly, adult pancreata from mice of desired genotype were dissected and fixed at 4°C overnight in 4% paraformaldehyde. Tissues were washed in PBS three-times and prepared as frozen sections, followed by immunofluorescence staining using mouse anti-Gαo described in [18]. Insulin co-staining was used to identify β-cells. DAPI co-staining was used to locate nuclei.

For insulin, E-cadherin, and tubulin staining, single cells or islets were used as shown in [16]. For single cells, islets were partially dissociated with trypsin, washed, overlaid onto human fibronectin-coated coverslips, and cultured overnight in RMPI-1640 media with FBS and 5.6 or 20 mM glucose. IF staining was then performed according to the following: cells or islets were extracted with methanol at -20°C for 5 minutes to remove free tubulin, followed by fixation with 4% paraformaldehyde (PFA) for 1 (for single cells)– 4 (for islets) hours, routine permeabilization and staining [16]. The primary antibodies used were: rabbit anti-α-Tubulin (Abcam, Cambridge, UK, #ab18251), purified mouse anti-E-Cadherin (BD, San Jose, CA, #610181), rabbit anti-Glu-tubulin (MilliporeSigma #AB3201), and guinea pig anti-insulin (Dako, Santa Clara, CA, #A0564). The mouse anti-Gαo antibody was described in [18]. Secondary antibodies are from Jackson ImmunoResearch (Alexa Fluor® 647 AffiniPure Donkey Anti-Guinea Pig IgG (H+L) (706-605-148); Alexa Fluor® 488 AffiniPure Donkey Anti-Rabbit IgG (H+L) (711-545-152), and Alexa Fluor® 594 AffiniPure Donkey Anti-Mouse (705-585-003). The dilution of all antibodies is 1:1000. Z-stacked images were captured using Nikon Eclipse A1R laser scanning confocal microscope. For super-resolution, images were captured at 0.125 μm intervals with DeltaVision OMX SIM Imaging System (GE technology) using a 60x NA1.4 lens and processed according to the manufacturer's instruction. Western blot follow routine biochemical methods.

## SG and MT density quantification

To quantify insulin subcellular localization (Fig 1), representative confocal images were taken from whole mount islets. A freehand line (~2 micrometer thick) was drawn around each cell-cell contact line. The insulin intensity underneath the line was then measured as the portion of total insulin. Image J was used for this assay.

MT density quantification using Image J follows the procedure described in [39]. Briefly, super-resolution MT images were captured as above at multiple z-depth. Line scans were then used to detect the size of the spaces between MTs by selecting 10 lines across the cell center with a 30-degree interval. A custom image J macro was used to create line selections and obtain the intensity profile. The lengths of regions without signal within the intensity profile were considered spaces between MTs, with the means and SEM presented.

## Statistics

For all studies, at least two biological repeats and two technical repeats were included (done at different days). Student t-test, two-tailed type II analysis, was used for comparisons between two groups of data. Two-way ANOVA with Holm-Sidak's multiple comparisons were used when more than two groups of data were compared. A $p$ value below 0.05 was considered significant in the text.

## Supporting information

**S1 Fig. Un-clipped western blot images.** This Fig shows the western-blot assay of Gαo protein in control and Gαo mutant islets. A portion of this images was used in Fig 4E.
(PDF)

**S1 Table. Raw numbers used in quantitative plots in all Figs.**
(XLSX)

## Author Contributions

**Conceptualization:** Irina Kaverina, Guoqiang Gu.

**Data curation:** Xiaodong Zhu, Kung-Hsien Ho, Irina Kaverina, Guoqiang Gu.

**Formal analysis:** Ruiying Hu, Xiaodong Zhu, Mingyang Yuan, Kung-Hsien Ho, Irina Kaverina, Guoqiang Gu.

**Funding acquisition:** Kung-Hsien Ho, Irina Kaverina, Guoqiang Gu.

**Investigation:** Xiaodong Zhu, Mingyang Yuan, Kung-Hsien Ho, Irina Kaverina, Guoqiang Gu.

**Methodology:** Ruiying Hu, Xiaodong Zhu, Mingyang Yuan, Kung-Hsien Ho, Guoqiang Gu.

**Project administration:** Irina Kaverina, Guoqiang Gu.

**Resources:** Guoqiang Gu.

**Software:** Ruiying Hu, Xiaodong Zhu, Mingyang Yuan.

**Supervision:** Ruiying Hu, Irina Kaverina, Guoqiang Gu.

**Writing – original draft:** Irina Kaverina, Guoqiang Gu.

**Writing – review & editing:** Ruiying Hu, Xiaodong Zhu, Mingyang Yuan, Kung-Hsien Ho, Irina Kaverina, Guoqiang Gu.

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
