## [Decision Letter · Decision Letter 0]

9 Nov 2020

PONE-D-20-33149

Microtubules and Gαo-signaling independently regulate the preferential secretion of newly synthesized insulin granules in pancreatic islet β cells

PLOS ONE

Dear Dr. Guoqiang Gu,

Thank you for submitting your manuscript to PLOS ONE. After careful consideration, we feel that it has merit but does not fully meet PLOS ONE’s publication criteria as it currently stands. Therefore, we invite you to submit a revised version of the manuscript that addresses the points raised during the review process.

We look forward to receiving your revised manuscript.

Kind regards,

Stephane Gasman

Academic Editor

PLOS ONE

Journal Requirements:

2.Thank you for stating the following in the Financial Disclosure section:

[This work was supported by grants from National Institutes of Diabetes and Digestive and Kidney Diseases (https://www.niddk.nih.gov/) DK106228 (to I.K. and G.G.), DK125696 and DK065949 (to G.G.), grants from National Institute of General Medical Sciences (https://www.nigms.nih.gov/) R35-GM127098 and R01-GM078373 (to I.K.). K.H. was supported by a postdoctoral fellowship from Eli Lilly and Company (LIFA fellowship 0101420) (https://www.amcp.org/resource-center/group-resources/residents-fellows/fellowships/Eli-Lilly). ]. 

We note that you received funding from a commercial source: [Eli Lilly and Company]

Additional Editor Comments (if provided):

Your article has been reviewed by two external reviewers. To warrant publication, the authors need to address the major concerns raised by both reviewers. Moreover, particular attention has to be given to avoid overstatement and overinterpretation of the results and methodology needs improvement to support the data.

Reviewers' comments:

Reviewer's Responses to Questions

**Comments to the Author**

1. Is the manuscript technically sound, and do the data support the conclusions?

Reviewer #1: Partly

Reviewer #2: Partly

2. Has the statistical analysis been performed appropriately and rigorously? 

Reviewer #1: Yes

Reviewer #2: No

3. Have the authors made all data underlying the findings in their manuscript fully available?

Reviewer #1: Yes

Reviewer #2: Yes

4. Is the manuscript presented in an intelligible fashion and written in standard English?

Reviewer #1: Yes

Reviewer #2: Yes

5. Review Comments to the Author

Reviewer #1: This paper dealing with insulin granule dynamics is a follow-up study from Du’s group of a recent paper on microtubule (ref 34) and an older study on Galphao (ref 15) and as such part the data partially recapitulate previous observations. The authors show here by using nocodazole treatment of mouse pancreatic islets or islets recovered from Galphao mouse that the intracellular distribution of insulin granules is modified. Following these observations, they further blocked protein synthesis with cycloheximide and conclude that older insulin granules are preferentially secreted when microtubules are disrupted. Although conceptually interesting the paper suffers from several weaknesses among which overstatement and overinterpretation of the results is a major issue. Indeed, in my opinion the methods used do allow to convincingly support the interpretation made on insulin granule age differences.

The description of the image analysis for Figure 1 fails to describe how the confocal plan was selected for each cell as this could largely affect the IG density whether the image is captured in the middle part of the cells or close to either top or bottom parts. Furthermore, the authors mention in the method section that the nuclear areas are avoided, but how are those areas defined is not mentioned. No nuclear staining seems to have been performed. Finally, it is not possible to identify the condition for which the values are statistically different in panel D/E.

The treatment of CHX is problematic as the authors did not should quantify its effect on novel granule biogenesis. Further the authors of ref 63 mention that this treatment actually specifically affects a short half-life protein important to ensure optimal exocytosis, suggesting that the present observation may not be specific to newly generated granules. According to this model why is there more newly synthesized granules released in the low glucose condition?

Glucose is missing in Figure 3D/E to indicate what the concentrations stand for!

The expression level of Galphao and the remaining peptide should be shown by western blot. As such with the limited data of Figure 4, the conclusion on line 287-289 “Thus, different pools of IGs were mobilized/secreted in response to Galphao inactivation and MT disassembly, implying that Galphao does not directly regulate MTs in betta cells” appears to be an overstatement at this time of the study as it is also possible that Galphao acts through a moderate MT depolymerization that is boosted by nocodazole treatment. Indeed, although not statistically significant there seem to be tendency that Galphao affected microtubule density and stability. Once more the data do not fully support the author’s conclusion.

Finally, another point that is at least worth discussing is the observation that Galphao was found associated with secretory granules membranes from different cell types including chromaffin cells. In these cells, it was shown that Galphao negatively regulates granule secretion by modulating cortical actin polymerization. The authors should discuss these findings and comment on the distribution of Galphao in pancreatic beta cells.

Reviewer #2: To the Authors

- The introduction is written well and puts the study appropriately in the context of the existing literature. At line 49 the authors state that beta cells have 10000 insulin granules citing Olofsson et al. - since there are other studies out there showing somewhat divergent numbers it would be appropriate to cite also these studies.

- In the results the authors show that nocodazole treatment leads to the expected disruption of microtubule filaments. When additionally treated with high glucose, insulin granules are only visible in the periphery ('docked') of the cells. This is shown by IF stainings and EM of islets (fig. 1 and 2). To measure SG density the drawing of lines is not ideal. The authors should surround the cells and calculate the SG density in the full area excluding the nucleus. Also, In the material and methods section, the authors state that (line 438-439) the 'lines were drawn to avoid the nuclei,..' however the cells in figure 1C seem to do exactly that. The authors should clarify this.

- The authors state at line 149 to having observed 'significant reduction in total IG levels...' as seen in figure 1D. To make a proper statement, the authors should additionally show the integrated values of their measurement, or ideally integrate the signal of the entire cell, since signal along the line may be heavily biased by where the line is set. The figure legend of figure 1 should state if the SD or the SEM is shown.

- Figure 2 replicates the data from figure 1 using electron micrographs. The authors state that the lack of microtubules following 12 h treatment with nocodazole (NOC) leads to a relative increase of IGs at the PM and respective reduction in the center of the cells. This effect is attributed to 'slow-but-detectable random IG movement' (line 155-156). First of all,

a major question is whether after 12 h nocodazole treatment the cells still secrete insulin. This long treatment may lead to also cause fragmentation of the Golgi and block insulin synthesis and IG biogenesis. This is especially relevant since the authors state before that there is a reduction of total insulin. Notably, in figure 2H the ER looks very stressed.

As a NOC treatment for 12 h seems to us very long, the authors should test whether granules and/or insulin are still properly generated under these conditions, or if the relative increase of PM IGs is the result of blocked biogenesis. In addition, MT depletion could also block the retrograde transport of IGs destined to degradation toward the perinuclear region, where endolysosomes are enriched. These two effects combined may account for IG depletion in the perinuclear region without confuting the role of MTs in promoting the centrifugal transport of newly generated IGs toward the PM.

- The authors then investigate the secretion of old IGs under conditions of blocked protein synthesis and/or lack of microtubules. To this aim beta cells were treated for 3 h with CHX and an additional hour with NOC. It is not clear why in figures 1&2 the NOC treatment was done for 12 h and here 1 h is sufficient to depolymerize MTs (figure 3). The authors should state the reasoning for this discrepancy in the protocols. The authors then measure the secreted insulin, assuming that 'newly synthesized insulin' should be secreted within 2-4 h. As expected, less insulin is secreted when protein biogenesis is blocked with CHX. Notably additional treatment with NOC further increased insulin secretion. The number of replicates is low and, therefore, the individual experiments should be shown with a dot plot (or similar; with SD rather than SEM). Non-parametric testing may be more suitable with low number of replicates.

- The authors go on to directly measure the secretion of old and new IGs. To this aim, islets were labelled overnight (12 h) with 3H-leucine, then chased for 2 h and then used for GSIS (about 2 h). Figure 3F shows the 'relative amount of radioactive insulin...' (line 241-242) to be lower for NOC vs DMSO samples. It is not readily clear from the text if these data are representing the medium or the islets and this should be better explained for clarification. Also here the individual values should be shown as bar graphs with SEM may not be representative.

- At line 254-256 the authors state that the 12 h window prior to the GSIS assay is considered to be new IGs. In this case, how can the authors make a statement about old IGs, if all the labelled IGs (see protocol above) should fall in the 12 h window of 'new IGs'? In other words, if 12 h old IGs are considered to be newly-synthesized, the authors would not detect old IGs using this approach. The authors should clarify this point.

- Next, the authors investigate a potential link between the MT-network and the G protein subunit Gao as the latter may regulate MTs. To address this, the authors replicate the data from figure 3 using a mutated Gao mouse line. They find increased secretion of 'old IGs' (similar to figure 3, the nomenclature needs to be clarified and data points shown in another format than bar graphs with SEM due to the low replication number) in isolated islets of the mutant mice upon high glucose. In figure 4F, the authors test the secretion of insulin in mutant vs control islets with or without NOC. At low glucose, the authors should also consistently add a statistical test, as it appears that NOC here reduces the secretion. With high glucose and potassium treatment, the insulin secretion is expectedly increased and is even further increased when NOC is included. However, the authors should test if there is a difference in secretion already between islets of mutant and control mice, to avoid falsely attributing the secretion difference between mutant and control islets to NOC, when there may already be an underlying effect in the mutation itself.

- Furthermore, the authors investigate a potential effect of mutant Gao on the MT network. Measuring MT density and stability, the authors could not detect any difference between Gao mutant and control islets.

- Finally, the authors show that NOC treatment induces a relative increase of PM IGs compared to IGs in the cell body. Using radiolabelling, the authors demonstrate that 12 h old IGs are increasingly released upon glucose stimulation and/or NOC treatment. Although Gao mutant islets appear to increase insulin secretion (total or 12 h old IGs), the effects appear to be additive, rather than related to the MT-mediated effects of NOC.

In general, data representation must be improved, for example using dot plots rather than bar graphs, especially where only low replicates are available. In these situations, non-parametric testing is superior to parametric testing (for example Mann-Whitney vs t-test).

The final paragraphs should speculate a bit more about the role of Gao to put it into context here, as the final result suggests no connection to MTs.

Minor

- It would be preferable to replace the term “IGs” to “insulin SGs”, since the latter more commonly used.

- the term "docked" SGs should be avoided whenever the analyzed images are obtained from fixed samples. In this instance one cannot indeed determine the residence time at the PM of the organelles, which may just be transient visitors. Moreover, “docking” implies a proximity <20 nm, i.e. below the resolution of fluorescence microscopy.

6. PLOS authors have the option to publish the peer review history of their article (what does this mean?). If published, this will include your full peer review and any attached files.

Reviewer #1: No

Reviewer #2: No

---

## [Author Response · Author response to Decision Letter 0]

30 Apr 2021

We thank the reviewers for constructive comments. All their critiques were now addressed, using new data and edits. The one-by-one rebuttal was included in the response letter.

---

## [Decision Letter · Decision Letter 1]

27 May 2021

PONE-D-20-33149R1

Microtubules and Gαo-signaling modulate the preferential secretion of young insulin secretory granules in islet β cells via independent pathways

PLOS ONE

Dear Dr. Gu,

Thank you for submitting your manuscript to PLOS ONE. The manuscript has been significantly improved. However, few minor concerns remain to be addressed to warrant publication. Therefore, we invite you to submit a revised version of the manuscript that addresses the points raised during the review process.

We look forward to receiving your revised manuscript.

Kind regards,

Stephane Gasman

Academic Editor

PLOS ONE

Journal Requirements:

Reviewers' comments:

Reviewer's Responses to Questions

**Comments to the Author**

1. If the authors have adequately addressed your comments raised in a previous round of review and you feel that this manuscript is now acceptable for publication, you may indicate that here to bypass the “Comments to the Author” section, enter your conflict of interest statement in the “Confidential to Editor” section, and submit your "Accept" recommendation.

Reviewer #1: All comments have been addressed

Reviewer #2: (No Response)

2. Is the manuscript technically sound, and do the data support the conclusions?

Reviewer #1: Yes

Reviewer #2: Partly

3. Has the statistical analysis been performed appropriately and rigorously? 

Reviewer #1: Yes

Reviewer #2: I Don't Know

4. Have the authors made all data underlying the findings in their manuscript fully available?

Reviewer #1: Yes

Reviewer #2: Yes

5. Is the manuscript presented in an intelligible fashion and written in standard English?

Reviewer #1: Yes

Reviewer #2: Yes

6. Review Comments to the Author

Reviewer #1: This paper dealing with insulin granule dynamics is a follow-up study from Du’s group of a recent paper on microtubule (ref 34) and an older study on Galphao (ref 15) and as such part the data partially recapitulate previous observations. The authors show here by using nocodazole treatment of mouse pancreatic islets or islets recovered from Galphao mouse that the intracellular distribution of insulin granules is modified. Following these observations, they further blocked protein synthesis with cycloheximide and conclude that older insulin granules are preferentially secreted when microtubules are disrupted.

In this revised version the authors have adequatly addressed all my concerns.

Reviewer #2: Hu and colleagues have made a considerable effort to address our original comments. Accordingly, the manuscript is now much clearer and significantly improved. There are however some aspects of it, which remain to be clarified.

Specific comments

Figure 1: The authors changed the quantification of insulin levels from confocal images according to our suggestions. However, now one can see that upon nocodazole treatment insulin levels are strongly reduced, while the percentage of insulin in the periphery is higher. What is the interpretation of these results?

Figure 2: How many times have these experiments been independently repeated? This figure could be visually more appealing and easier to interpret if the plasma membrane would be marked, for example with arrows or a coloured outline.

Figure 3: The radiolabeling measurements provide information only about the secretion of the young secretory granules, meaning that conclusions about the behaviour of the old secretory granules remain speculative.

Notably, cycloheximide treatment did not significantly affect insulin secretion. A limitation of this approach, which would need to be acknowledged, is that each stimulation induces only the secretion of a small fraction of the granule stores. Thus, one would need to repeat the cycloheximide treatment multiple times (which is however incompatible with cell viability) in order to reliably assess the contribution of insulin biosynthesis to secretion.

Figure 4: As in figure 3, the claims about the old secretory granules are speculative. As we requested, the graphs now show the single values, and it seems there are many extreme values that make the comparison significant.

Line 195: "... movement of SGs to endosomes for degradation." – the authors should probably refer to “lysosomes” rather than to “endosomes”

7. PLOS authors have the option to publish the peer review history of their article (what does this mean?). If published, this will include your full peer review and any attached files.

Reviewer #1: No

Reviewer #2: No

---

## [Author Response · Author response to Decision Letter 1]

9 Jun 2021

We thank both reviewers for their comments to improve the manuscript. All comments are addressed in the re-revised version. Please see details in the attached rebuttal file.

---

## [Editor Report · Decision Letter 2]

16 Jun 2021

Microtubules and Gαo-signaling modulate the preferential secretion of young insulin secretory granules in islet β cells via independent pathways

PONE-D-20-33149R2

Dear Dr. Guoqiang Gu,

We’re pleased to inform you that your manuscript has been judged scientifically suitable for publication and will be formally accepted for publication once it meets all outstanding technical requirements.

Kind regards,

Stephane Gasman

Academic Editor

PLOS ONE
---

## [Editor Report · Acceptance letter]

13 Jul 2021

PONE-D-20-33149R2 

Microtubules and Gαo-signaling modulate the preferential secretion of young insulin secretory granules in islet β cells via independent pathways 

Dear Dr. Gu:

I'm pleased to inform you that your manuscript has been deemed suitable for publication in PLOS ONE. Congratulations! Your manuscript is now with our production department. 

Kind regards, 

on behalf of

Dr. Stephane Gasman 

Academic Editor

PLOS ONE